# InSe as a case between 3D and 2D layered crystals for excitons

T.V. Shubina[1], W. Desrat [2], M. Moret[2], A. Tiberj[2], O. Briot[2], V.Yu. Davydov [1], A.V. Platonov [1], M.A. Semina[1] & B. Gil[1,2]

InSe is a promising material in many aspects where the role of excitons is decisive. Here we report the sequential appearance in its luminescence of the exciton, the biexciton, and the P-band of the exciton-exciton scattering while the excitation power increases. The strict energy and momentum conservation rules of the P-band are used to reexamine the exciton binding energy. The new value $\geq 20$ meV is markedly higher than the currently accepted one (14 meV), being however well consistent with the robustness of the excitons up to room temperature. A peak controlled by the Sommerfeld factor is found near the bandgap (~1.36 eV). Our findings supported by theoretical calculations taking into account the anisotropic material parameters question the pure three-dimensional character of the exciton in InSe, assumed up to now. The refined character and parameters of the exciton are of paramount importance for the successful application of InSe in nanophotonics.

[1] Ioffe Institute, 26 Politekhnicheskaya, St Petersburg 194021, Russia. [2] Laboratoire Charles Coulomb (L2C), Université de Montpellier, CNRS, Montpellier FR-34095, France. Correspondence and requests for materials should be addressed to T.V.S. (email: shubina@beam.ioffe.ru) or to W.D. (email: wilfried.desrat@umontpellier.fr)

Mono-chalcogenides are the layered compounds that were in great demand in the last century for basic research into the physics of highly anisotropic semi-conductors[1–3], and that have recently attracted great interest as parent materials for two-dimensional (2D) crystals[4]. A bright representative of this family Indium Selenide obeys high electron mobility, attractive quantum physics, tunable absorption, and extremely strong photoresponse[5–7]. The absorption edge of InSe, being close to that in silicon, offers sufficient solar energy conversion[8,9]. Owing to these exceptional properties, InSe is now widely considered as a seminal compound for a number of potential applications that are transformative in many fields of modern physics and nanotechnology[10].

Many interesting optical phenomena in semiconductors are associated with free excitons and exciton complexes formed due to the Coulomb interaction between particles. For instance, the hybrid biexcitons that can involve states from different valleys[11,12], and the scattering-induced dark biexcitons[13], were discovered in transition metal dichalcogenides. Similar studies of mono-chalcogenides are at the very beginning, although it is generally accepted that exciton and exciton complexes will determine their use in nanophotonics.

The III–VI mono-chalcogenides (III = Ga, In; VI = S, Se, and Te) consist of tetralayers type Se-In(Ga)-In(Ga)-Se bound by weak van der Waals forces. The stacking sequence of tetralayers leads to different polytypes (see Supplementary Note 1). Both widely used $\gamma$ and $\beta$ polytypes of InSe have direct energy gap[14,15]. In a single-tetralayer limit, the type of the band structure changes for the indirect with the Mexican hat-like valence band dispersion. In contrast to GaSe and GaS, this transition in InSe can occur gradually on a dozen of tetralayers[16,17]. Although the exciton transitions between the uppermost valence band and the lowermost conduction band are direct, they are fully allowed only for light polarization $E\|c$, where c is the direction normal to the layers. The transitions with $E\perp c$ in InSe occur possible through a weak spin–orbit interaction[18–20] (see Supplementary Note 2).

In 1968, Andriyashik et al.[21] detected two exciton peaks in the absorption spectrum of bulk InSe, which they ascribed to the ground and first excited states. They derived the direct energy gap $E_g \simeq 1.36$ eV and the exciton binding energy $R_X \sim 37$ meV. Ten years later, Camassel et al.[22] have revised these data using similar transmission measurements. The energies of three observed peaks led to $E_g = 1.353$ eV and $R_X = 14.5$ meV. Even smaller values were obtained by differential magneto-optical measurements[23]. The modelings in all these experimental data were done using the three-dimensional (3D) theory of allowed direct excitonic transitions. Note that the agreement with this model for the excited states was insufficient, in contrast to GaSe[24,25] (see Supplementary Note 4). This, as well as the possible restrictions imposed by the selection rules for exciton states, have been neglected. The partial applicability of this model was a crucial argument in favor of the 3D character of excitons in InSe, instead of the expected 2D for this layered crystal.

Here, it is worth mentioning about some factors which can influence absorption spectra, such as the Sommerfeld factor which provides a peak at an absorption edge due to the effects of excitons within the continuum[3], and the interference peaks which can appear in the region of relatively low absorption above the ground exciton energy. Nevertheless, the 1978's values are frequently used up to now, and the 3D exciton case is commonly accepted.

As an alternative way for the determination of these fundamental parameters, we propose considering the nonlinear photoluminescence (PL) processes. In general, with increasing excitation power, one should successively observe the appearance of the PL lines of the exciton (X), biexciton (M), and the so-called P-band of exciton–exciton (X−X) scattering. In the latter, one of the exciton is scattered into a photon, while the other is scattered into a first excited state ($P_2$) or to the continuum ($P_\infty$)[26]. Such series have been observed in many semiconductors; however, up to now this was never reported for the mono-chalcogenides. Importantly, the energies of emitting photons are strictly dependent on the exciton binding energy as shown in the scheme in Fig. 1a, which depicts both constituents of P-band—$P_2$ and $P_\infty$.

The X transitions exhibit usually a linear dependence of PL intensity on excitation power. In contrast to that, both biexciton recombination cascade and X–X scattering are characterized by a quadratic power dependence. To confirm the biexciton formation, a superlinear dependence of the PL intensity on power is enough. For the inelastic scattering between two excitons, the exact quadratic dependence is typical because this process dominates over others at high excitation power. At the end, this X–X scattering results in the emergence of stimulated emission (SE)[27].

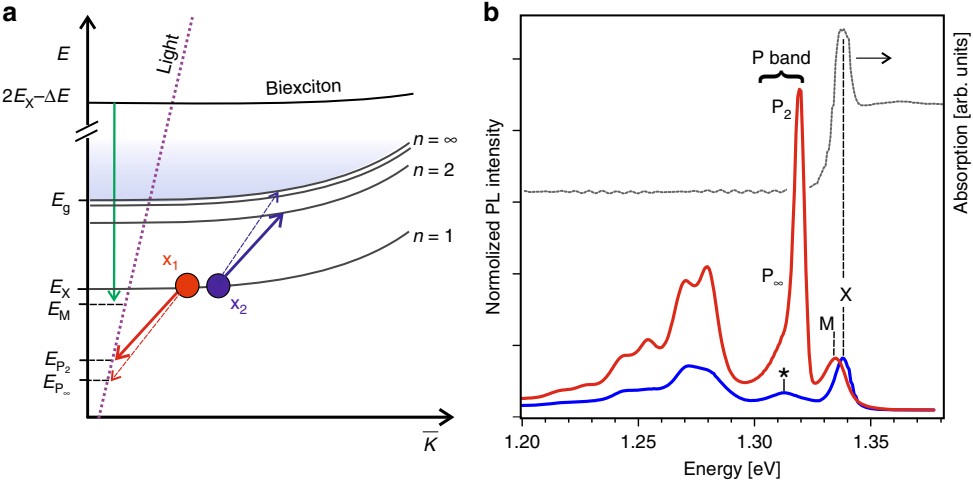

**Fig. 1** Exciton, biexciton, and exciton–exciton scattering. **a** Schematic representation of the biexciton decay and the X–X scattering which involves the $n = 2$ and $n = \infty$ states shown by solid and dashed arrows, respectively. The exciton-level notation is given for the 3D case; it should start from $n = 0$ for the 2D case. The characteristic energies are marked on the vertical axis ($\Delta E \approx E_X - E_M$). **b** Photoluminescence spectra (linear scale) measured under low (50 Wcm$^{-2}$, blue line) and high (0.3 MWcm$^{-2}$, red line) excitation powers in undoped InSe. An absorption spectrum (gray line) is shown for comparison

Formally, the SE may be asserted when and only when the PL intensity shows a superquadratic dependence on power.

A feature related to the biexciton was observed in GaSe, about 2 meV below the X line, by means of the nonlinear 2D Fourier transform technique[28]. A similar study of InSe did not report on such an observation[29]. At power-dependent PL studies, both biexcitonic and P bands were revealed in layered lead iodide crystals[30], whereas only P and SE bands were reported for bulk GaSe[31,32] and InSe[33]. The superquadratic dependence expected for SE was published for GaSe only, where it likely involves the direct states situated somewhat above the indirect ones. The band structure of InSe which is a highly ansiotropic semiconductor with a valence band splitting of hundreds meV[18] is hardly consistent with such a scenario. However, it makes this compound suitable for the investigations of excitons and exciton complexes, as well as the exciton character, missed in the previous studies.

Here, we demonstrate the perfect series of X, M, and P emission bands in InSe, which obey the theoretically predicted laws. We use the strict energy and momentum conservation rules of the P-band constituents to determine the exciton-binding energy and estimate the bandgap width. The comparison of the obtained parameters with the calculated ones, using an anisotropic model for an exciton, as well as taking into account the temperature robustness of excitons and the appearance of the Sommerfeld peak in absorption, cast serious doubts on the purely 3D character of the exciton in InSe.

## Results

**Photoluminescence**. The optical experiments were carried out on samples of good structural quality freshly cleaved from bulk InSe grown by the Bridgman–Stockbarger process. We investigated two kinds of samples: without intentional doping (naturally n-type), called further as undoped, and p-type Zn-doped InSe. Details are given in the Methods section.

Figure 1b presents the PL spectra of undoped InSe measured at $T = 10$ K with low and high excitation powers. For the sake of demonstration, these spectra are normalized to the maximal intensities of the peaks marked as X(M). The defect-related bands observed below 1.3 eV are specified in the Supplementary Method 4. Here, we focus on the shorter-wavelength lines with the maxima at 1.338 eV (X), 1.335 eV (M), and 1.320 eV (P-band). The line at 1.338 eV perfectly corresponds to the free exciton recombination[22,23,34]. It is also well matching the ground exciton peak detected in the absorption spectrum of a thin sample. Its well-defined energy ($E_X$) is the starting point for further analysis.

In our samples, the M and X peaks cannot be separately resolved because of their closeness and broadening. However, we notice that the joint X + M peak shifts by ~3 meV to the lower energy with increasing the excitation power (Fig. 2a). The band renormalization cannot explain this behavior since the energy of the nearest P-band is very constant, while the scattering to higher-energy states should be very sensitive to this process. This joint peak exhibits a superlinear dependence on pump power with an exponent $k = 1.3$ (Fig. 2b). We assume that it is the result of the admixture of a biexciton (although a trion contribution cannot be completely excluded). The superlinear growth occurs up to the threshold of intense X–X scattering. Beyond that the exponent decreases down to $k = 0.7$. Note that the competition between these two processes – biexciton formation and X–X scattering – is always in favor of the latter[35].

The attribution of the 1.320 eV band to the X–X scattering P-band is proved by its power dependence shown in Fig. 2b. The P-band comprises two overlapping components, $P_2$ and $P_\infty$. They

could be well separated due to a significant energy distance between the first excited state and a free-particle bandgap corresponding to the maximum of the $P_\infty$ line (see Fig. 1a). In the PL spectrum measured at $T = 10$ K, the basic peak of the P-band is situated 18 meV below X. The principal question is which of the components, $P_2$ or $P_\infty$, produce the maximum of the P-band. For GaSe, the dominant peak was ascribed to $P_2$[31]. We also incline to this option based on its lineshape. At low temperature, the $P_2$ should display the Lorentzian lineshape, because the scattering has the well-defined level $n = 2$ (in 3D notation) as a final state. On the contrary, the $P_\infty$ line should be widened due to the large number of possible final states. The higher-energy part of the P-band in Fig. 1b can be perfectly fitted by a Lorentzian, while its lower-energy part deviates markedly from that. From this we can conclude that the difference $(E_X - E_{P_2}) = 18$ MeV in our case.

The determination of the $P_\infty$ energy in undoped InSe (naturally n-type) is complicated by the closeness of the line marked by (*) in Fig. 1b, which was reported in many previous studies as a donor-related transition[34,36]. To get rid of this problem, we have investigated a Zn-doped InSe sample, where the possible donors are compensated by the p-type impurity. Respectively, the (*)-line disappears. Spectra measured in this sample contain only the peaks of both components (Fig. 2c) separated by ~55 meV. These peaks exist up to the onset of the exciton–electron scattering, well defined by means of a P-band shift and the deviation of the power dependence from the quadratic law (Fig. 2d).

**Temperature dependencies**. Figure 3 shows the PL spectra measured under high pumping power in a wide temperature range. Whereas all defect-related lines quench fast at the temperature of ~60 K, both the exciton-related emission and P-band survive almost up to room temperature. This would correspond to an exciton-binding energy of about 20 meV. These emission bands are significantly broadened at temperatures above 80 K, which indicates the involvement of phonons to the recombination process. The increase of the gap between the X and P lines (inset of Fig. 3) is related to the gradually increasing contributions of the exciton–electron scattering and electron–hole plasma recombination to the formation of the P-band[27].

The absorption spectra measured in a thin (40 μm) undoped InSe sample exhibit also the X peak surviving up to high temperatures (≥ 250 K). Details on the absorption spectra fitting using the Urbach rule, shown in Supplementary Fig. 6, are given in the Supplementary Method 5. Despite our best efforts in measuring samples of different thicknesses, we have failed to find the definite peaks of excited exciton levels as reported in ref. [22], which is the almost unique description in the literature. Instead of that, a smooth peak (hump) was recorded ~25 meV above the X peak. The intensity of this peak is much lower than that of X, but it is well visible in an enlarged scale (see Fig. 4a). Its shift and quenching with temperature follow those of the ground exciton peak (Fig. 4b).

## Discussion and conclusion

The radiative decay of free excitons takes place only when their momentum is located inside a rather narrow light cone, which results in the existence of a bottleneck. This leads to the accumulation of excitons, which stimulates the enhanced exciton–exciton scattering. In InSe, this situation is aggravated by the fact that the lowest in energy direct exciton state is weakly allowed with E⊥c. Thus, the scattering of excitons is a very suitable effect for determining the binding energy of the excitons.

In the 3D case, the states of the Wannier–Mott exciton obey a 3D hydrogen-like dependence $E_n = -R_X/n^2$[37] (counting from $E_g$),

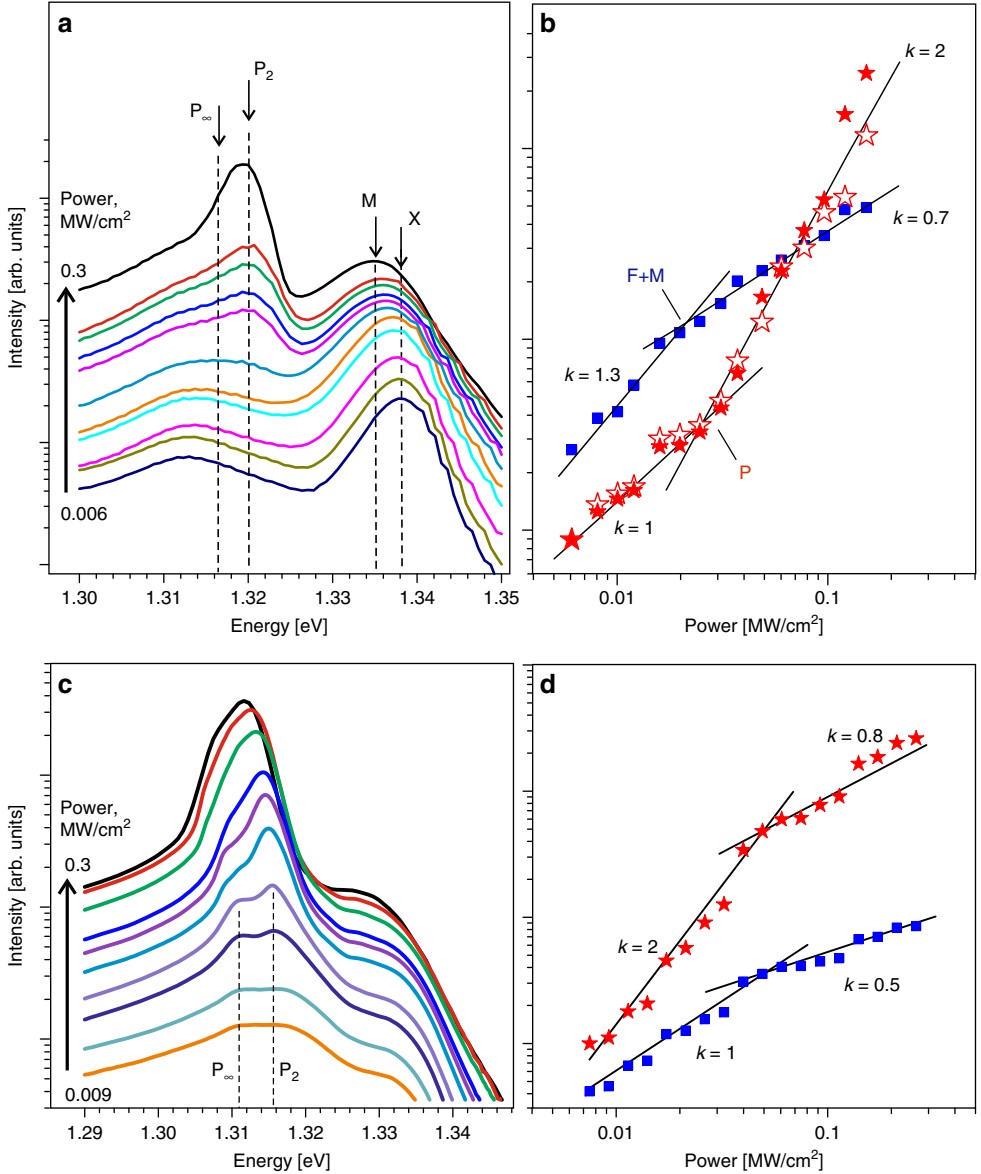

**Fig. 2** Excitation power dependence of photoluminescence. **a**, **c** Selected PL spectra measured at different excitation powers (log scale) in undoped (**a**) and doped (**c**) InSe samples. **b**, **d** Power dependencies of integral PL intensity shown for the joint X + M peak (blue squares) and P-band (red stars) in the undoped (**b**) and doped (**d**) InSe. Black lines show fittings with the exponents marked nearby. In (**b**), solid and open red stars present the P-band data obtained, respectively, as for a whole band and as a sum of two components

where $R_X$ is the exciton binding energy, while $n = 1, 2, 3...$ are the ground and excited exciton levels. For an ideal isotropic crystal, the energies of the P-band constituents are expressed as[26]

$$E^{3D} = E_g(T) - R_X\left(2 - \frac{1}{n^2}\right) - 3\delta k_B T, \quad n = 1, 2, 3... \quad (1)$$

For scattering associated with the $n = 2$ level when $E^{3D} = E_{P_2}$, Eq. (1) can be expressed via $E_X - E_{P_2}$, i.e., via the difference between the ground exciton energy ($E_X = E_g - R_X$) and the energy of the $P_2$ band (see Fig. 1a). At low power and temperature when the last term in Eq. (1) is negligible, we straightway obtain $(E_X - E_{P_2})^{3D} = \frac{3}{4}R_X$. For $E_X - E_{P_2} = 18\,\text{MeV}$ derived from our photoluminescence experiments, $R_X = 24\,\text{meV}$ and $E_g = 1.362\,\text{eV}$. The possible correction by the term $3\delta k_B T$ is insignificant because the PL maximum does not markedly shift either with increasing power, as would be expected in the case of additional laser heating, or with temperature in the range $10 \div 20$ K. Among different points on the sample surface, the energies of

the PL lines vary within ±0.5 meV (see Supplementary Fig. 5). Thus the estimated error in the $R_X$ determination is about ±1 meV.

In the Zn-doped InSe sample, we have measured an energy difference $E_{P_2} - E_{P_\infty} \sim 5\,\text{meV}$. This value does not depend on temperature at all. For the 3D hydrogen-like case, it leads to $R_X \simeq 20\,\text{meV}$ which may be considered as a lower limit of the exciton Rydberg energy in InSe. Note that $E_g \sim 1.36\,\text{eV}$ is in agreement with the published experimental data dispersed in a wide range[38,39] and with our experimental absorption data as well. We also underline the closeness of the Rydberg values obtained in different InSe samples, which vary around the average value of 22 meV.

Let us now consider the ideal 2D case. For allowed transitions, the energies of bound excitons obey the dependence $E_n = -R_X/(n + 1/2)^2$, where $n = 0, 1, 2, \ldots$ and the ground-state corresponds to $n = 0$[40]. As can be easily counted, the $P_2$ scattering will produce a photon with energy $\frac{32}{9}R_X$, and the bandgap will be situated at the

energy $4R_X$, i.e., ~20 meV above $E_X$. Forbidden transitions in the 2D case can start from $n = 1$ only[1,40]. The respective bandgap would be $4R_X/9 = 28$ meV above $E_X$. We discard this possibility because the transitions in InSe are at least partly allowed.

We have marked in Fig. 4 the values of direct $E_g$ predicted for the 3D and 2D ideal cases. They are close to each other, and both are situated below the maximum of the discovered minor peak. Note that the Coulomb interaction not only ensures the formation of a series of discrete levels below the ionization edge but also changes the wave functions of the continuum states above this

edge[40]. It leads to an absorption peak controlled by the Sommerfeld factor which writes $C(j) = 2/(j + 1/2)^3$ with $j = 0 \dots \infty$. The contribution of the $j$th continuum excitonic state (unbound exciton) to the absorption drops exponentially with its increasing detuning from the edge. As a result, the Sommerfeld peak appears very close to the bandgap energy. The most pronounced peak occurs in 2D confinement where it can locally increase the step-like 2D absorption by a factor of 2. To form such a peak, the exciton states must be allowed[1,3] which is fulfilled in InSe. Thus, the clear diagnostic of the Sommerfeld peak is one of the ways to determine $E_g$.

The value of the exciton-binding energy $R_X \approx 20$ meV, obtained in this work, is noticeably higher than the previously reported around 14 meV in assumption of 3D exciton character[22,23]. In this connection, it should be noted that one can reliably conclude on the 3D exciton character if and only if there is a consistent series of excited states. This requirement was not fulfilled in previous experiments where the inconsistency in derived $R_X$ values approached 40%. The analysis of the published data is given in Supplementary Note 4 and Supplementary Table 3.

Furthermore, let us consider the selection rules of excitonic transitions depicted in Fig. 5a (see Supplementary Note 2 and Supplementary Fig. 1). For $E \perp c$ that is the typical configuration for absorption measurements, the ground exciton state of $s$-symmetry is partly allowed via the weak spin–orbit interaction[18,19,41]. The excited exciton states $2s$ and $3s$ of the same symmetry have oscillator strength, which is markedly smaller than would be expected for fully allowed excited transitions which obey either $n^{-3}$ or $(2n-1)^{-3}$ laws for the hydrogen-like 3D or 2D excitons[3,40]. The $2p$ excited state, which is fully forbidden in conventional absorption measurements (allowed in two-photons ones) can be mixed with the closely situated $3s$ state[41,42] that will hardly enhance the probability of its observation.

To illustrate the strong impact of the selection rules on optical processes, we have measured the PL spectra from a sample facet using two polarizations: $E \perp c$ and $E \parallel c$. We observe the dramatic enhancement of PL intensity with $E \parallel c$ in full agreement with the selection rules in InSe (see Supplementary Fig. 2). Systematic studies of many InSe samples carried out independently in two of our research centers have not shown any signs of excited exciton states in the spectra of both absorption and reflectivity (not presented here). Thus we can conclude that the excited states can hardly be reliably recorded in InSe, because of the symmetry-related limitations resulting in the rather large linewidth and the small binding energy of these states.

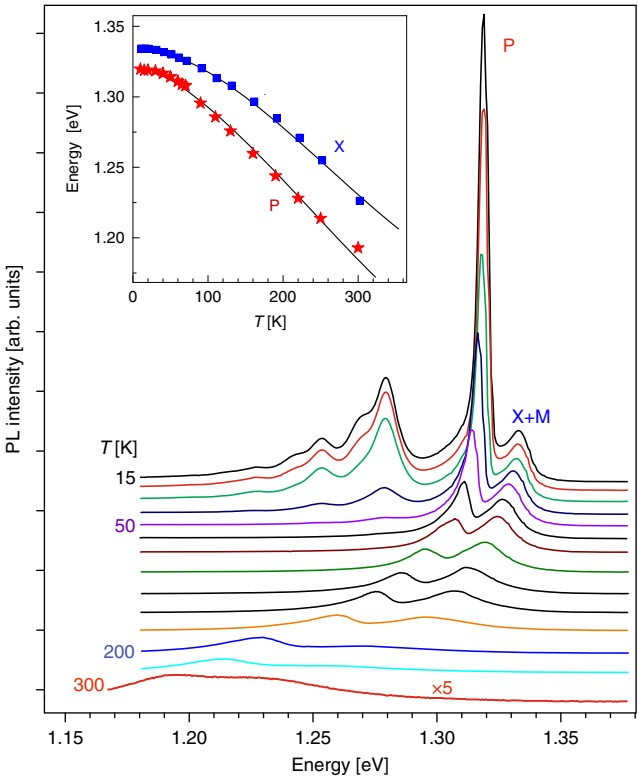

**Fig. 3** Temperature dependence of photoluminescence. PL spectra measured at high excitation power in the undoped InSe sample at different temperatures (linear scale). The inset presents the dependencies of the X and P energies vs. temperature

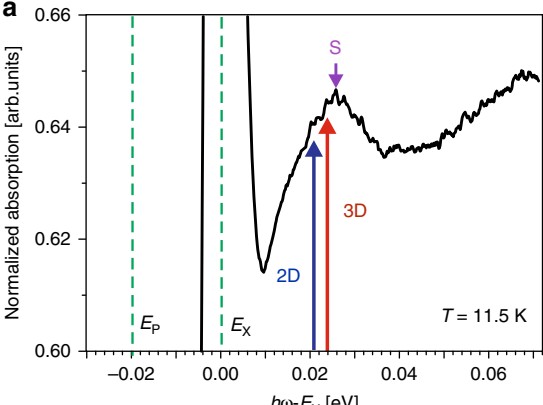
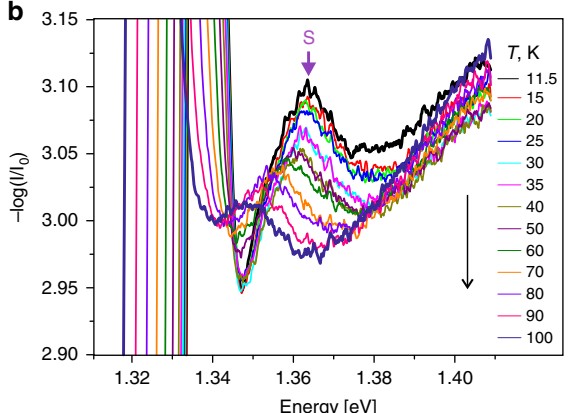

**Fig. 4** Sommerfeld peak. **a** An absorption spectrum measured at 11.5 K normalized to the intensity of the X peak, whose energy is taken as zero. The vertical axis scale is chosen to focus on the Sommerfeld peak (S). The bandgap energies derived from the P-band analysis are shown for the 3D (red arrow) and 2D allowed (blue arrow) cases. The peaks of excited states are not resolved. **b** The absorption coefficient measured at different temperatures

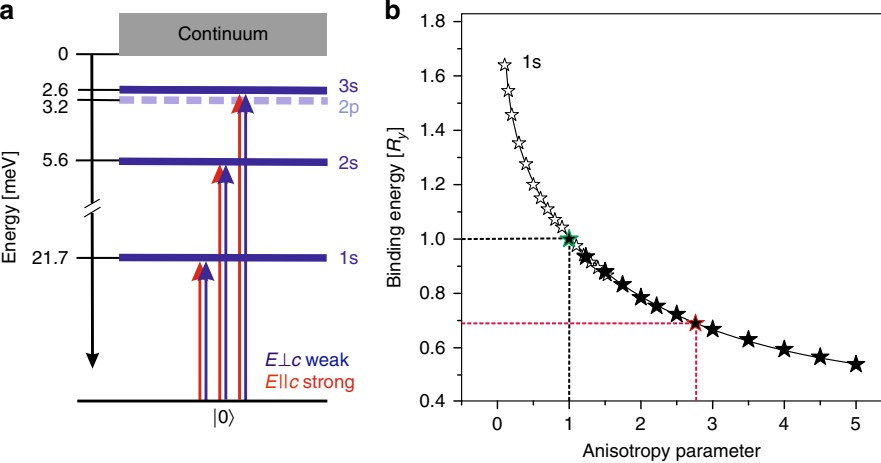

**Fig. 5** Excitons in anisotropic InSe. **a** Schematic of the exciton levels in InSe, where the calculated binding energies are plotted from the ionization edge. **b** Calculated dependence of the 1s exciton-binding energy vs. the parameter of anisotropy. Our data, marked by the full symbols, coincide well with the data reported in ref. [42], marked by the empty symbols. The green and red lines indicate the ideal isotropic and realistic anisotropic InSe cases, respectively

The $R_X$ magnitudes which we have derived from the scattering experiments for the ideal 2D and 3D cases are close to each other, which hampers doing the choice between them. In GaSe with similar $R_X \approx 20$ meV, the exciton was treated as 3D because of the out-of-plane expansion of the wave functions, induced by the contribution of Se orbitals[24]. In the highly anisotropic InSe, this effect might be also pronounced[43].

Here, it is worth mentioning that both 3D and 2D hydrogen-like cases are the solutions of the same Schrödinger equation for uniaxial crystals in the effective mass approximation[44]. Its solution depend on the anisotropy factor $\chi = \kappa_\perp \mu_\perp / \kappa_\parallel \mu_\parallel$, where $\mu_{\perp,\parallel}$ and $\kappa_{\perp,\parallel}$ are, respectively, the reduced masses and relative dielectric constants in the directions normal ($\perp$) or parallel ($\parallel$) to the $c$-axis. If $\chi = 1$, the solution corresponds to the 3D case, while $\chi = 0$ leads to the 2D case[42,45]. We have performed theoretical modeling as described in Supplementary Note 3 and Supplementary Tables 1 and 2. The found binding energies for the ground and excited exciton states are depicted in Fig. 5a. Using realistic parameters obtained by alternative experimental methods[15,46,47], namely $\mu_\perp/m_0 = 0.118$, $\mu_\parallel/m_0 = 0.055$, $\kappa_\perp = 8$, and $\kappa_\parallel = 6.4$, we have got the binding energy for 1s state of 21.7 meV, which is very close to our average experimental value $R_X \simeq 22$ meV.

Figure 5b presents the calculated dependence of the exciton-binding energy 1s on the anisotropy parameter $\chi$, which agrees well with the general dependence reported in ref. [42]. For the anisotropy parameter greater than unity, as in InSe, the exciton-binding energy is less than in an ideal isotropic crystal (by ~30% in InSe). Such anisotropy pushes the exciton from the range between the ideal cases $\chi = 1$ and $\chi = 0$. We emphasize that in order to obtain $\chi = 1$, the parameters of such isotropic InSe should differ greatly from the real anisotropic parameters reliably determined by various methods. Our research shows that the exciton physics in InSe is far from trivial. It requires to combine the prolate exciton wave function, elongated along the $c$-axis due to the smaller reduced mass in this direction, with a large reduced mass in the $x,y$-plane, which leads to a high binding energy and temperature robustness. Such a state resembles in part a one-dimensional exciton.

As for the physical background of observed phenomena, it is worth recalling the long-predicted small dispersion of excitons in InSe, which is due to its high ionicity[48]. Recently, it was confirmed by electron energy-loss spectroscopy (EELS), which showed that the band structure near the 1.3-eV exciton is flat in the distance from the zone center[15]. In such conditions, a hole effective mass and a reduced mass $\mu$ are heavier[20,49], and the value $R_X \propto \mu$ is higher than with the parabolic dispersion. In the presence of stacking disorder, which confines excitons within a finite number of tetralayers[50], the band structure is further modified. These factors, along with the strong anisotropy in dielectric constants in $k$-space[1], do not give a chance for the exciton states in the naturally anisotropic InSe to match neither a perfect $1/n^2$ 3D series nor the one described as $1/(n + 1/2)^2$ for extreme two-dimensionality in their commonly used meanings. Additional theoretical studies are clearly needed to elucidate the excitonics in InSe.

In conclusion, we have detected the successive appearance of the free exciton, biexciton, and P-band in InSe using PL measurements with increasing excitation power up to ~0.3 MW cm$^{-2}$. The well-defined energies of the P-band constituents with respect to the X peak allow us to estimate the values of the exciton-binding energy and bandgap energy. Thus we have proposed and proven a novel method for the determination of fundamental parameters in such semiconductors. It is worth noting that our data are obtained using rather transparent modeling which can be easily checked. We assume that the difference between the previously published data and ours occurs not only due to the limited adaptation of ideal models, developed for isotropic 3D and 2D excitons, but also due to the existing uncertainty in the real band structure and exciton wave functions in some layered crystals, such the Indium Selenide considered here. In particular, the adequate results cannot be obtained neglecting the symmetry and anisotropy effects in such compounds. Our findings including the enhanced stability of the exciton in the high temperature range and the first observation of the exciton complex in InSe are of paramount importance for future applications of this compound in nanophotonics and quantum optics.

## Methods

**Structural characterization**. The structural quality of the samples was characterized by X-ray diffraction and Raman studies. Both methods confirmed the good quality of the studied samples. The experimental results are presented in Supplementary Methods 1 and 2 and Supplementary Figs. 3 and 4.

**Optical measurements**. The cleaved samples (millimeter-sized parallelepipeds) were mounted on the cold finger of a closed cycle helium cryostat with the possibility of temperature variation in the 10–350 K range. A continuous 15 mW red laser (650 nm) was used to measure low-intensity PL. A Q-switched green laser

(532 nm) was used for high power pumping. The pump intensity was varied using neutral filters. The PL was detected by a Hamamatsu InGaAs photomultiplier cooled at 77 K. Besides PL, we measured transmission in thinner (40–70 μm) InSe samples with a halogen lamp from $T = 10$ K up to 350 K. Other details are given in the Supplementary Method 3.

## Data availability

The data sets analyzed during this study are available from the corresponding author on reasonable request.

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

## Acknowledgements

T.V.S., M.A.S, V.Yu.D., A.V.P., and B.G. acknowledge the support of the Government of the Russian Federation (Project no. 14.W03.31.0011 at the Ioffe Institute). T.V.S. and M.A.S also acknowledge the support of theoretical studies by the Russian Science Foundation (project # 19-12-00273). We thank M.M. Glazov for very fruitful discussions.

## Author contributions

B.G. initiated the study. The optical experiments were carried out by W.D., M.M., A.T. A.V.P. and O.B. The X-ray diffraction was performed by M.M., the Raman studies by V.Yu.D., the theoretical consideration by M.A.S. and T.V.S. The general data analysis was done by T.V.S. All co-authors discussed the data. T.V.S. and W.D. wrote the paper.

## Additional information

**Competing interests:** The authors declare no competing interests.

