## [Peer Review File · Nature Communications]

Reviewers' comments:

Reviewer #1 (Remarks to the Author):

In their paper, Desrat et al. report on the detection of the free exciton, biexciton, and P band in InSe using PL measurements with increasing excitation power.

The authors demonstrate a new value ≥ 20 meV, higher than the currently accepted 14 meV, which is however consistent with the robustness of excitons up to room temperature.

The manuscript is well written and the discussion scientifically sounds. The authors have provided a careful and accurate work, that however lacks of broad interest. These results are interesting only for the community working on InSe. Thus, manuscript deserves publication on a specialist journal on layered materials.

Therefore, it is surely unsuitable for Nature Communications, while it could be a good contribution for 2D Materials, Physical Review B or Scientific Reports.

Reviewer #2 (Remarks to the Author):

The manuscript of W. Desrat et al. reports on a study of excitonic properties of InSe layered crystal, which successfully combines experiment and modelling.

This is a very timely study on InSe layered crystal, a semiconductor material that has been recently gaining unprecedented interest thanks to its unique band structure, high electron mobility and tuneable absorption. Because of these interesting properties, InSe is now widely considered as a promising platform for a number of potential applications that are transformative not only within the large family of vdW materials, but also with respect to conventional semiconductors, such as silicon.

The authors reexamined the exciton binding energy in beta-polytype InSe and reported a new value of more than 20 meV, which is significantly higher than the currently accepted one, i.e. 14 meV. The obtained value is consistent with the robustness of excitons up to room temperature. A peak governed by the Sommerfeld factor was found near the bandgap of about 1.36 eV. These results are

transformative in terms of understanding the character of the excitons in InSe, which can't be considered as purely three dimensional any more.

The overall impression of the work is very good. It is a thorough study, and there is a good agreement between the experimental data and the modelling. The findings are very important not only from the fundamental point of view, but also from the practical one, as understanding of the excitonic properties of InSe are crucial for its application in nanophotonics. In my opinion, it is timely and of significant interest to the wide audience of Nature Communication. Therefore, I recommend to publish the manuscript.

Reviewer #3 (Remarks to the Author):

Authors study the exciton, biexcitons, and P band of exciton-exciton scattering in indium selenide (InSe) as one of a layered

monochalcogenide materials. They claim that for the first time they observe these series in monochalcogenides. The study is interesting however there are several comments which are mentioned in below:

1- Part of the information in the manuscript is not clearly

described. For example, the parameters in most equations such as

Eqs. 1 and 2 are not defined. At low power and temperature, Eq. 1 is converted to Eq. 2 where E_g is changed to E_X without defining them. It is not clear that where the 18 meV used in Eq. 2 comes from.

2- Authors claim that the peak marked by * in Fig. 1b is related to a donor-like defects which is not straightforward. Extra measurement or evidence is required to prove this. In the supplemental information authors mention that the * peak is not related to the transitions between a donor level and valance band because of the band structure however they do not provide any calculated band structure of InSe to clarify this.

3- For ideal 2D case, authors are using an equation for the energies of bound excitons which I think they have used the equation from the literature but they are not citing any reference.

4- Whole Section ``InSe exciton between the 3D and 2D cases' is not clear. For example authors mentioned that the ratio $R_M/R=0.15$ where R is not defined. It is not clear how these ratio show the intermediate case between 2D and 3D. They also find that their exciton binding energy is significantly higher than that reported in the literature but do not clarify the reason. They also mention that the exciton Rydberg should be higher but do not mention with respect to what? I assume that μ and m_0 in the exciton Rydberg formula are the reduced mass and the free electron mass but it is not defined in the manuscript.

5- Authors mention that they have characterized the structural quality of their samples by Raman studied but there is not Raman measurement provided.

6- In supplemental information, authors assign the A, B, and C peaks in Fig. 1b to donor-acceptor transition, donor-to-band transition and exciton bound to donor levels according to the literature. However it is more convincing if they provide their own measurements to prove these.

7- The different color PL spectra in Fig. 2 are not defined.

In general, the manuscript is not written in a clear form and even there are not enough theoretical or experimental support to prove the claims. The major correction is required and I recommend the authors to submit their manuscript to another journal.

Response to the referees' comments

Reviewer #1 (Remarks to the Author):

In their paper, Desrat et al. report on the detection of the free exciton, biexciton, and P band in InSe using PL measurements with increasing excitation power. The authors demonstrate a new value ≥ 20 meV, higher than the currently accepted 14 meV, which is however consistent with the robustness of excitons up to room temperature. The manuscript is well written and the discussion scientifically sounds. The authors have provided a careful and accurate work, that however lacks of broad interest. These results are interesting only for the community working on InSe. Thus, manuscript deserves publication on a specialist journal on layered materials. Therefore, it is surely unsuitable for Nature Communications, while it could be a good contribution for 2D Materials, Physical Review B or Scientific Reports. The authors have provided a careful and accurate work, that however lacks of broad interest. These results are interesting only for the community working on InSe.

We thank Reviewer for the positive evaluation of our paper as a careful and accurate work with interesting results. The comment on possibly limited audience has stimulated us to rewrite partly the manuscript. In particular, the new first paragraph underlines the significance of InSe for both classical semiconductor physics and the physics of 2D structures. Indium selenide has a lot of common features with other layered semiconductors like GaSe. Thus its study has a crucial importance for the III-VI family as a whole. The global significance of our paper for the physics of semiconductors lays also in a novel method of the determination of semiconductor fundamental parameters via the inelastic scattering effect. This is underlined now in conclusion. Owing this changes our paper will attract the interest of wide audience.

Basic actions:

1) The new paragraph (the first in Introduction) with new references is added:

“Mono-chalcogenides are the layered compounds that were in great demand in the last century for basic research into the physics of highly anisotropic semiconductors [1-3] and that have recently attracted great interest as parent materials for two-dimensional (2D) crystals [4]. A bright representative of this family – the Indium Selenide obeys high electron mobility, attractive quantum physics, tunable absorption, and extremely strong photoresponse [5-7]. The absorption edge of InSe, being close to that in silicon, offers sufficient solar energy conversion [8-9]. Owing to these exceptional properties, InSe is now widely considered as a seminal compound for a number of potential applications that are transformative in many fields of modern physics and nanotechnology [10].”

2) In Conclusion we write:

Actually, we have proposed and proven a novel method for the determination of fundamental parameters in such semiconductors.

3) Accordingly, minor modifications are done in the abstract.

Reviewer #2 (Remarks to the Author):

The manuscript of W. Desrat et al. reports on a study of excitonic properties of InSe layered crystal, which successfully combines experiment and modelling.

This is a very timely study on InSe layered crystal, a semiconductor material that has been recently gaining unprecedented interest thanks to its unique band structure, high electron mobility and tuneable absorption. Because of these interesting properties, InSe is now widely considered as a promising platform for a number of potential applications that are transformative not only within the large family of vdW materials, but also with respect to conventional semiconductors, such as silicon.

The authors reexamined the exciton binding energy in beta-polytype InSe and reported a new value of more than 20 meV, which is significantly higher than the currently accepted one, i.e. 14 meV. The obtained value is consistent with the robustness of excitons up to room temperature. A peak governed by the Sommerfeld factor was found near the bandgap of about 1.36 eV. These results are transformative in terms of understanding the character of the excitons in InSe, which can't be considered as purely three dimensional any more.

The overall impression of the work is very good. It is a thorough study, and there is a good agreement between the experimental data and the modelling. The findings are very important not only from the fundamental point of view, but also from the practical one, as understanding of the excitonic properties of InSe are crucial for its application in nanophotonics. In my opinion, it is timely and of significant interest to the wide audience of Nature Communication. Therefore, I recommend to publish the manuscript.

We highly appreciate the positive evaluation of our paper. Actually, Reviewer presents the evidences of global significance of indium selenite, which have helped us a lot in our work in modification of the paper.

Reviewer #3 (Remarks to the Author):

Authors study the exciton, biexcitons, and P band of exciton-exciton scattering in indium selenide (InSe) as one of a layered monochalcogenide materials. They claim that for the first time they observe these series in monochalcogenides. The study is interesting however there are several comments which are mentioned in below:

We thank Reviewer for the careful reading of our paper, accurate and concise formulation of main results, and a lot of very useful remarks. They help us to improve significantly the paper and exclude both serious drawbacks and misprints. Below are our answers on the particular remarks.

1- Part of the information in the manuscript is not clearly described. For example, the parameters in most equations such as Eqs. 1 and 2 are not defined. At low power and temperature, Eq. 1 is converted to Eq. 2 where E_g is changed to E_X without defining them. It is not clear that where the 18 meV used in Eq. 2 comes from.

Basic action:

1) This part of the text is rewritten as:

“For scattering associated with $n = 2$ level when $E^{3D} = E_{P2}$, Eq. (1) can be expressed via $E_X - E_{P2}$, i.e. via the difference between the ground exciton energy ($E_X = E_g - R_X$) and the energy of the P_2 band (see Fig. 1a). ... For $E_X - E_{P2} = 18$ meV derived from our photoluminescence experiments...”

2) In the section “Photoluminescence”, we note:

“From this we can conclude that the difference $E_X - E_{P2} = 18$ meV in our case”

2- Authors claim that the peak marked by * in Fig. 1b is related to a donor-like defects which is not straightforward. Extra measurement or evidence is required to prove this. In the supplemental information authors mention that the * peak is not related to the transitions between a donor level and valance band because of the band structure however they do not provide any calculated band structure of InSe to clarify this.

It is obvious misunderstanding. We have claimed that the line cannot be related to the indirect bandgap transitions, which should be much wider and weaker at low temperature as we observed. Based on literature data, we have explicitly written that the line marked by (*) is a donor-related transition, as reported in many previous studies [24,25].

This assignment is supported by our distinct observation of its quenching in the p-doped structures with respect to the naturally n-doped InSe. Besides, the line intensity saturates with the power increase and quench fast with the temperature rise. Such a behavior is typical for the defect-related emission. We believe that these data are sufficient to characterize this line, which, by the way, is not a subject of our study.

Basic actions:

1) *The comments on defect related lines are given in SM Section V “Defect-related emission” and partly in the main text.*

2) *The schematic band structure enough for our consideration is presented in Supplementary materials (SM) Section I.*

3- *For ideal 2D case, authors are using an equation for the energies of bound excitons which I think they have used the equation from the literature but they are not citing any reference.*

The needed reference is added (the new Ref. [41]).

4- Whole Section 'InSe exciton between the 3D and 2D cases' is not clear. For example authors mentioned that the ratio $R_M/R=0.15$ where R is not defined. It is not clear how these ratio show the intermediate case between 2D and 3D.

We have changed significantly this section and renamed it for accuracy as “Excitons in naturally anisotropic InSe”. We add the consideration of selection rules and the results of theoretical calculations of exciton states in the anisotropic InSe. The scheme of excitonic transitions with calculated binding energies of the ground and excited states, as well as the general dependence of 1s state on the anisotropy parameter is given in new Figure 5.

Concerning the presented example - this is obvious misprint; it should be R_M/R_X . However, we agree that the intermediate value R_M/R_X between 1 and 2 of biexciton is not obligatory related to the discussed peculiarities of the exciton in InSe and exclude that along with similar non-informative sentences.

Basic actions.

- 1) Modified Section “Excitons in naturally anisotropic InSe” (former “InSe exciton between the 3D and 2D cases”) with the extended consideration and new figure (see below).
- 2) Non-informative phrases have been excluded as not important for our discussion and putting unnecessary questions.

They also find that their exciton binding energy is significantly higher than that reported in the literature but do not clarify the reason.

Possible reasons were previously briefly discussed in the introduction. Now we present extended explanation in the Discussion in Section “Excitons in naturally anisotropic InSe” (former Section “InSe exciton between the 3D and 2D cases”). Note that the small values were only reported by the same research group of Camassel et al.; hence, it could be a systematic misunderstanding of the complicated problem. This group did not pay proper attention the discrepancy in R_X values derived from the energy gaps $\Delta E_{2,1}$ and $\Delta E_{3,2}$ (see our analysis in the SM section II). Besides, the treatment of all previous measurements neglected the selection rules of excitonic transitions that strongly influence the absorption measurements. To demonstrate their importance, we present for the first time the PL measurements from the crystal facet in two polarizations (see Fig. 2 in the SM Section I). Finally, previous studies neglected the strong anisotropy of all parameters, which dramatically modifies the exciton wave functions.

Basic actions.

The modified paragraphs are:

“The value of the exciton binding energy $R_X \approx 20$ meV, obtained in this work, is noticeably higher than the previously reported around 14 meV in assumption of 3D exciton character [22,23]. In this connection, it should be noted that one can reliably

conclude on the 3D exciton character if then and only if there is a consistent series of excited states. This requirement was not fulfilled in previous experiments where the inconsistency in derived R_X values approached 40%. (Analysis of published data is given in SM Section II.)

Further; let us consider the selection rules of excitonic transitions depicted in Fig. 5a (see SM Section I for details). For $E \perp c$ that is the typical configuration for absorption measurements, the ground exciton state of s-symmetry is partly allowed via the weak spin-orbit interaction [18,19,42]. The excited exciton states 2s and 3s of the same symmetry have oscillator strength which is markedly smaller than would be expected for fully allowed excited transitions which obey either n^{-3} or $(2n-1)^{-3}$ laws for the hydrogen-like 3D or 2D excitons [3,41]. The 2p excited state, which is fully forbidden in conventional absorption measurements (allowed in two-photons ones) can be mixed with the closely situated 3s-state [42,42] that will hardly enhance the probability of its observation.”

The new paragraph is added

“To illustrate the strong impact of the selection rules on optical processes, we have measured the PL spectra from a sample facet using two polarizations: $E \perp c$ and $E \parallel c$. We observe the dramatic enhancement of PL intensity with $E \parallel c$ in full agreement with the selection rules in InSe (see Fig. 2, SM Section I). Systematic studies of many InSe samples carried out independently in two of our research centers have not shown any signs of excited exciton states in the spectra of both absorption and reflectivity (not presented here). Thus we can conclude that the excited states can hardly be reliably recorded in InSe, because of the symmetry-related limitations resulting in the small binding energy and rather large linewidth of these states.”

We have also extended the phrases in introduction, paragraph 4:

“Note that the agreement with this model for the excited states was insufficient, in contrast to GaSe [24,25] (see SM Section II for our analysis). This, as well as the possible restrictions imposed by the selection rules for exciton states, have been neglected.”

They also mention that the exciton Rydberg should be higher but do not mention with respect to what? I assume that μ and m_0 in the exciton Rydberg formula are the reduced mass and the free electron mass but it is not defined in the manuscript.

Thank you for pointing this uncertainty. We complete this sentence by “is higher than with the parabolic dispersion” and introduce the term “reduced mass” in the previous paragraph.

5- Authors mention that they have characterized the structural quality of their samples by Raman studied but there is not Raman measurement provided.

We present the perfect Raman spectra in the SM Section III. These spectra are measured in a wide frequency range to show the low-frequency modes rigid layer modes, because only they allow establishing the type of folding. To the best of our knowledge, such spectra was not published previously for InSe.

6- In supplemental information, authors assign the A, B, and C peaks in Fig. 1b to donor-acceptor transition, donor-to-band transition and exciton bound to donor levels according to the literature. However it is more convincing if they provide their own measurements to prove these.

We have supplied needed information to confirm their defect origin. SM Section V includes the analysis of literature data and the confirmation of their defect origin by our temperature and power measurements. Note that the study of defect-related emission bands below 1.3 eV is out of scope of this paper and we have decided to do not attract unnecessary attention to them by excluding their marking in Fig. 1b.

Basic action: *The data on defect-related lines are given in SM Section V; symbols A-C removed from Fig. 1b.*

7- The different color PL spectra in Fig. 2 are not defined.

We have introduced the power bar with boundary values to the Fig. 2 (a) and (c).

In general, the manuscript is not written in a clear form and even there are not enough theoretical or experimental support to prove the claims.

We did a great job to make the text much clear. We add new data that is enough for experimental and theoretical support to prove our concept. It includes theoretical modelling of binding energies of different excitonic states accounting the anisotropy and much extended explanation of obtained results, as well as the unique data on polarized PL and Raman scattering in a wide frequency range.

Basic actions.

1) The new paragraphs on the theoretical investigation and the discussion of obtained results, which include Fig. 5a have been added to the main text:

“Here it is worth mentioning that both 3D and 2D cases are the solutions of the same Schrödinger equation for uniaxial crystals in effective mass approximation [45]. Its solution depends on the anisotropy factor $\chi = \kappa_{\perp} \mu_{\perp} / \kappa_{\parallel} \mu_{\parallel}$, where $\mu_{\perp, \parallel}$ and $\kappa_{\perp, \parallel}$ are, respectively, the reduced masses and relative dielectric constants in the directions normal (\perp) or parallel (\parallel) to the c-axis. If $\chi=1$, the solution corresponds to 3D case, while $\chi=0$ leads to the 2D case [3,43,46]. We have performed theoretical calculations as described in SM Section I. The found binding energies for the ground and excited exciton states are depicted in Fig. 5a. Using realistic parameters obtained by alternative experimental methods [15, 47, 48], namely $\mu_{\perp}/m_0 = 0.118$, $\mu_{\parallel}/$

$m_0 = 0.055$, $\kappa_{\perp} = 8$, and $\kappa_{\parallel} = 6.4$, we have got the binding energy for $1s$ state of 21.7 meV, which is very close to our average experimental value $R_x \approx 22$ meV.

Figure~5b presents the calculated dependence of the $s1$ exciton binding energy on the anisotropy parameter χ , which it is well consistent with the general dependence reported in Ref. [43]. For the anisotropy parameter greater than unity, as in InSe, the exciton binding energy is less than in an ideal isotropic crystal (by $\sim 30\%$ in InSe). Such anisotropy pushes the exciton from the range between the ideal cases $\chi=1$ and $\chi=0$. We emphasize that in order to obtain $\chi = 1$, the parameters of such isotropic InSe should differ greatly from the real anisotropic parameters reliably determined by various methods. Our research shows that the exciton physics in InSe is far from trivial. It requires to combine the prolate exciton wave function, elongated along the z axis due to the smaller reduced mass in this direction, with a large reduced mass in the x,y -plane, which leads to a high binding energy and temperature robustness. Such a state resembles in part a one-dimensional exciton.

2) Supplementary Materials (SM) are significantly extended; now they comprise 6 sections. The Section I includes new subsections: “Band structure of InSe”, “Selection rules of optical transitions”, “Anisotropic model for excitons in InSe”. Section II is “Analysis of published data on excitons in InSe”. Section III “Samples characterization” includes the subsection “Raman studies”. The SM also contains the Section IV “Optical measurements”, extended Section V “Defect-related emission”, and Section VI “Temperature dependent measurements”.

3) Extended explanation of observed phenomena is given in the main text.

4) 15 new references are added.

3) The text of the manuscript is corrected to exclude repeating places and unnecessary information. Overall English polishing has been done.

The major correction is required and I recommend the authors to submit their manuscript to another journal.

We believe that by the performed corrections and supplying all information required by the Reviewer we significantly improve this manuscript. The global significance of our results is not only for the InSe - perspective material for optoelectronics and nanophotonics, but also for general physics of semiconductors because our knowledge of excitons in strongly anisotropic layered compounds is significantly enhanced. Based on these advantages we resubmit our paper to Nature Communications.

REVIEWERS' COMMENTS:

Reviewer #3 (Remarks to the Author):

In the revised manuscript, Shubina et al has significantly modified their manuscript. They have done Raman measurements to classify InSe as beta type. They have also added more discussion about the qualitatively comparison of their experiment with theory in the manuscript as well as in the supplemental information which makes the discussion more strong. The manuscript and the supplemental information now is well written and I think that would be of interest to people who are working on inhomogeneous materials so I recommend to publish the manuscript.